# Intraovarian, Isoform-Specific Transcriptional Roles of Progesterone Receptor in Ovulation

**DOI:** 10.3390/cells11091563

**Published:** 2022-05-05

**Authors:** Kirsten M. Smith, Doan T. Dinh, Lisa K. Akison, Matilda Nicholls, Kylie R. Dunning, Atsushi Morimoto, John P. Lydon, Darryl L. Russell, Rebecca L. Robker

**Affiliations:** 1Robinson Research Institute, School of Biomedicine, University of Adelaide, Adelaide 5000, Australia; kirsten.smith@adelaide.edu.au (K.M.S.); doan.dinh@adelaide.edu.au (D.T.D.); matildanicholls11@gmail.com (M.N.); kylie.dunning@adelaide.edu.au (K.R.D.); ats.japan.0930@gmail.com (A.M.); darryl.russell@adelaide.edu.au (D.L.R.); 2School of Biomedical Sciences, University of Queensland, St. Lucia 4072, Australia; l.akison@uq.edu.au; 3Department of Molecular & Cellular Biology, Baylor College of Medicine, Houston, TX 77030, USA; jlydon@bcm.edu

**Keywords:** progesterone receptor, PGR-A, PGR-B, PRKO, transcriptome, ovulation, ovarian stroma, granulosa cells, pathways analysis, respiration

## Abstract

Progesterone receptor (PGR) activity is obligatory for mammalian ovulation; however, there is no established direct functional pathway explaining how progesterone receptor completely and specifically regulates oocyte release. This study examined the overarching cell- and isoform-specific effects of the PGR within each cellular compartment of the ovary, using mice null for the PGR (PRKO), as well as isoform-specific null mice. The PGR was expressed in ovarian granulosa and stromal cells and although PRKO ovaries showed no visible histological changes in preovulatory ovarian morphology, follicle rupture did not occur. Reciprocal ovarian transplant experiments established the necessity of ovarian PGR expression for ovulation. Cumulus–oocyte complexes of PRKO mice exhibited normal morphology but showed some altered gene expression. The examination of mitochondrial activity showed subtle differences in PRKO oocytes but no differences in granulosa cell respiration, glycolysis or β-oxidation. Concurrently, RNA-seq identified novel functional pathways through which the PGR may regulate ovulation. PGR-A was the predominant transcriptionally active isoform in granulosa cells and 154 key PGR-dependent genes were identified, including a secondary network of transcription factors. In addition, the PGR regulated unique gene networks in the ovarian stroma. Collectively, we establish the effector pathways activated by the PGR across the ovarian cell types and conclude that PGR coordinates gene expression in the cumulus, granulosa and stromal cells at ovulation. Identifying these networks linking the PGR to ovulation provides novel targets for fertility therapeutics and nonhormonal contraceptive development.

## 1. Introduction

Ovulation is a highly coordinated, multifaceted series of events that culminate in the release of the mature oocyte. This is a pivotal event in reproduction and identifying the critical mechanisms that lead to oocyte release is essential to our basic understanding of female fertility and the development of novel therapeutic interventions for humans. Following the LH surge, substantial energy-intensive remodeling occurs within the ovary. Within the preovulatory follicle, the granulosa cells coordinate the release of prostaglandins and other paracrine signaling factors to initiate the processes that mediate follicular rupture [1,2,3,4]. These factors mediate the cumulus–oocyte complex (COC) expansion, resumption of oocyte maturation and induce a cumulus cell invasive phenotype for the extrusion of the oocyte [5,6,7,8,9]. The granulosa cells themselves also release proteases to remodel the extracellular matrix and follicular architecture [10,11,12,13,14]. Surrounding the follicle, in the stromal compartment of the ovary, ovulation involves an influx of immune cells, vascular remodeling and smooth muscle contraction [15,16,17,18,19,20,21,22]. Much of what is known about the control of ovulation has been investigated within the ovarian follicle. However, the stromal tissue of the ovary is emerging as a similarly critical regulator of ovarian function [23]. In particular, we recently identified that stromal collagen deposition and optimal mitochondrial function are key contributors to successful ovulation (unpublished data). Therefore, it is important to consider stromal gene expression to fully understand the mechanisms required for ovulation. Isolating the critical molecular pathways and genes for these functions will provide novel targets for the development of nonhormonal contraceptives and inform the treatment of female infertility.

The progesterone receptor (PGR) is well established as a key mediator of ovulation, with PGR activity being an absolute requisite for follicular rupture across multiple species [24,25,26,27,28]. The two main isoforms of the progesterone receptor PGR-A and PGR-B are present at tissue-specific, tightly controlled ratios to transcriptionally regulate functions across the entire female reproductive system [29]. PGR knockout (PRKO) mice which are null for both isoforms display a number of reproductive defects including impaired uterine decidualization, severe underdevelopment of mammary glands, the absence of sexual receptivity behavior and critically, a compete block in oocyte release even after exogenous superovulation treatment [24,28,30,31,32]. The PGR is induced 750-fold in mouse granulosa cells following the LH surge and regulates the expression of important ovulatory genes including *Adamts1*, *Edn2* and *Pparg* [14,28,33,34,35,36,37]. Importantly, however, no known PGR-regulated gene individually explains the requirement of PGR for follicle rupture, as null mouse models for each of these genes still have some ability to ovulate, in stark contrast to the complete block in ovulation seen in PRKO mice. Our previous low-density array of the whole PRKO ovary at 10 h post-hCG and a microarray analysis of PRKO granulosa cells at 8 h post-hCG identified a number of novel PGR-regulated genes [34,38], with recent high-throughput sequencing analysis further demonstrating the extent to which PGR-A and PGR-B isoforms regulate the ovulatory transcriptional landscape [39]. Despite this, there is no established direct pathway explaining how the progesterone receptor regulates ovulation in such a profoundly complete and specific manner. Additionally, since mice null for the PGR-A isoform (PRAKO) have an anovulation phenotype while PGR-B null mice (PRBKO) ovulate normally [39,40,41], there are isoform-specific transcriptional roles of the PGR required for follicle rupture that remain to be identified.

This study holistically examined the overarching, cell- and isoform-specific effects of the PGR within the ovary to understand PGR-regulated mechanisms that explain its necessity for ovulation. Using mice null for the PGR (PRKO), as well as isoform-specific PRAKO and PRBKO mice, we comprehensively examined each cellular component of the ovary to identify functional pathways through which PGR may regulate ovulation. Ultimately, RNA-seq was used to determine the isoform-specific effect of the PGR on transcriptional regulation in both the granulosa cells and the ovarian stroma to identify novel downstream pathways functionally linking PGR to ovulation.

## 2. Materials and Methods

### 2.1. Animals and Superovulation

This research was approved by the University of Adelaide Animal Ethics Committee (ethics approval numbers m/2018/100 and m/2018/122) and conducted in accordance with the National Health and Medical Research Council (NHMRC) Australian Code of Practice for the Care and Use of Animals for Scientific Purposes. Mice were housed in a 12 h light:12 h dark housing conditions with rodent chow and water provided ad libitum.

Female CBA x C57BL/6 F1 (CBAF1) mice were obtained from Laboratory Animal Services (University of Adelaide). Genetically modified mice null for both PGR isoforms (PRKO; Pgr^tm1Bwo^), null specifically for the PGR-A isoform (PRAKO; Pgr^tm1Omc^) or null for the PGR-B isoform only (PRBKO; Pgr^tm2Omc^) were obtained from Jackson Laboratory (Bar Harbor, ME, USA) [24,40,41]. A second KO model null for all PGR isoforms (Pgr^tm1Lyd^) was used in some experiments where indicated. In this strain, designated PRlacZ, a neomycin cassette and lacZ insertion disrupt the expression of the PGR gene [42]. Mice were double genotyped using ear biopsies to assign experiments then confirmed using tail biopsies collected at time of experiments.

Female mice at 21–25 days old (unless otherwise specified) were hormonally stimulated to synchronously induce folliculogenesis and ovulation. Mice were injected intraperitoneally with 5 IU equine chorionic gonadotropin (eCG). Then, after 44 h, 5 IU human chorionic gonadotropin (hCG) was administered by intraperitoneal injection. Mice were humanely killed by cervical dislocation at specified timepoints post-hCG injection and the ovarian material was collected.

### 2.2. Histology and Immunofluorescence

Ovaries were fixed in cold 4% paraformaldehyde at 4 °C for 24–48 h. Ovaries were embedded in paraffin wax and sectioned at 5 µm thickness.

For histological assessment, sections from PRlacZ strain were stained with hematoxylin and eosin (H&E) and imaged using a Nanozoomer digital slide scanner with a 40× objective.

For immunofluorescence, sections were dewaxed in xylene and rehydrated; then, heat-induced antigen retrieval was performed using citrate buffer (pH 6). Sections were blocked in solution of 1% BSA/9% normal goat serum (NGS) in tris-buffered saline (TBS) for 1 h at RT. This was followed by incubation with primary antibody against PGR (1:500, D8Q2J, cat no. 8757, Cell Signaling, Danvers, MA, USA) diluted in 1% NGS-TBS overnight at 4 °C. Slides were washed with 0.025% Triton X-100-TBS, then incubated with secondary antibodies (1:2000) and Hoechst 33,342 (1:500) diluted in 1% NGS-TBS for 1 h at RT. Images were acquired using an Olympus FV3000 confocal microscope (Olympus, Tokyo, Japan).

### 2.3. Ovarian Transplant

Donor ovaries were obtained from 3.5–6.5-week-old WT and KO females from the PRlacZ colony that were humanely killed by cervical dislocation and ovaries removed and kept in PBS at room temperature. WT females from the PRlacZ strain were used as recipient mice at 3–4 months old. Recipient mice were anesthetized using isoflurane inhalation and a small mid-dorsal incision used to access the ovary. The ovarian fat pad was clamped with a serrefine clamp to secure the ovary in place. Ovarian transplants were performed as described by Nagy et al. [43]. Briefly, a small incision in the bursa on the opposite side to the oviduct was used to slip the ovary out of the bursa. The ovary was then removed by pinching off the supporting stalk with fine forceps. The donor ovary was then inserted into the recipient’s bursal sac. The ovarian tissues were returned to the body cavity before repeating on the opposite side. After the ovarian transplant was complete the wounds were closed using wound clips. Recipient females were placed on a heating pad to maintain body temperature during anesthesia recovery.

For histological assessment, a WT and a KO donor ovary were transplanted onto the contralateral sides of the same recipient females (*n* = 3). At 3 to 6 weeks following surgery, recipient mice were stimulated with eCG followed by hCG and humanely killed at 23 h post-hCG. Ovaries were fixed and the entire ovaries sectioned and stained with H&E (see above) for histological examination. A range of 23–43 sections per ovary were assessed for the number of anovulatory follicles. Both unruptured follicles not yet luteinized with expanded COCs and luteinized, unruptured follicles with entrapped oocytes were included in the definition of anovulatory follicles.

For the fertility study, only one (either WT or KO) ovary was transplanted into the ovariectomized recipient females (*n* = 3 per donor genotype). At two-week post-surgery, the recipient females were housed with WT males (PRlacZ strain) for three months and vaginal plugs assessed for evidence of copulation. Resulting pups were genotyped as previously described [44] to confirm ovary of origin.

### 2.4. Oocyte JC-1 Staining

COCs were collected at 10 h post-hCG and denuded with 50 μL hyaluronidase (1000 IU/mL, Invitrogen Australia, Mulgrave, Vic, AU) in collection media (αMEM-HEPES). Denuded oocytes were washed in fresh collection media then incubated in media containing 6 μM JC-1 dye (GIBCO, Invitrogen Australia). Oocytes were incubated for 15 min at 37 °C to allow JC-1 aggregates to form in high-membrane potential mitochondria (red fluorescence). Oocytes were then immediately imaged through an optical cross-section at the widest point of the oocyte using the Fluoview FV10i Olympus confocal microscope for green fluorescence emission (525 nm, low membrane potential) and red fluorescence (590 nm, high membrane potential) using a 60× objective. A total of 58 oocytes from 4 KO mice and 61 oocytes from 4 heterozygous littermates were imaged (PRlacZ strain). Using AnalySIS LS professional software (Olympus Australia, Mt Waverly, Vic, AU), a rectangle was placed across the oocyte image and the average green or red fluorescence intensity in each pixel column (0.3 μm intervals) across the box was determined. The mean and SE of the red or green fluorescence was calculated across all PRKO or PGR heterozygous (PRHet) oocytes. Statistical significance was determined by a mixed-model two-way ANOVA.

### 2.5. COC Microarray

COCs were collected by needle puncture from antral follicles needle of KO and heterozygous (PRlacZ) ovaries at 8 h post-hCG, then snap-frozen in liquid nitrogen until use. Heterozygous PGR-null mice were used as a control as ovulation rates are comparable to wildtype mice [39,42]. A total of 4 biological replicates per genotype were obtained, each consisting of COCs pooled from 3 mice. RNA was extracted using the TRIzol extraction method and DNAse-treated before use. Microarray analysis was performed by the Adelaide Microarray Centre (AMC, SA, Australia). Samples RNA integrity thresholds over 8.0 were verified using an Agilent bioanalyzer (Agilent Technologies, Santa Clara, CA, USA). Sample preparation was performed by AMC using Affymetrix GeneChip kits (Affymetrix, Santa Clara, CA, USA) according to the manufacturer’s instructions. Samples were hybridized overnight to Affymetrix GeneChip Mouse Gene 1.0 ST arrays (Affymetrix, Santa Clara, CA, USA). Assays were washed, stained with fluorescently labelled antibody then scanned using the Affymetrix GeneChip Scanner 3000 7G plus.

Data were normalized by fitting to the linear model and differential gene expression between PRKO and PRHet COCs determined using limma-voom. LogFC > 1 and Padj < 0.05 criteria were applied to obtain a list of differentially expressed genes (DEGs). Raw data from the COC microarray are available on GEO (GSE92436). COC gene expression was compared to granulosa cell (GC) gene expression, using previously published adjacent microarray data of PRKO 8 h GCs [34,38]. Data were obtained from GEO (GSE92437) and analyzed using the method described above.

### 2.6. Seahorse Extracellular Flux Assay

Ovaries were collected from female mice (PRKO, PRAKO and PRBKO strains) at 10 h post-hCG and granulosa cells were collected from antral follicles via puncturing. Cells were separated by the addition of 1 IU of hyaluronidase per 1 mL of handling media for 5 min and pipetted through a fine glass pipette. Granulosa cells were then purified through a 70 µm filter to remove oocytes and any undigested tissue. Cells were seeded at 150,000 live cells per well in Seahorse cell culture plates that had been coated with fibronectin overnight.

Mitochondrial stress test (Agilent, Santa Clara, CA, USA) was performed as per the manufacturer’s instructions with cell-optimized drug concentrations. In brief, cells were seeded into mitochondrial media (1 mM pyruvate, 10 mM glucose, 2 mM glutamine and 15 mg/L phenol red in DMEM-based medium (Sigma Aldrich, pH 7.4 at 37 °C), incubated at 37 °C for 15 min, then the oxygen consumption rate (OCR) measured on the Agilent Seahorse XFe96 extracellular flux analyzer. The Seahorse injection plate was hydrated overnight prior to the assay being performed and then calibrated, and cells were sequentially treated with oligomycin (1 µM), carbonyl cyanide 4-(trifluoromethoxy)phenylhydrazone (FCCP, 2.5 µM) and a combination of antimycin A (1 µM) and rotenone (1 µM).

The glycolysis stress test (Agilent) was performed as above but using glycolysis media (2 mM glutamine and 15 mg/L phenol red in DMEM-based medium (Sigma Aldrich), pH 7.4 at 37 °C). Extracellular acidification rate (ECAR) was measured on the Seahorse XFe96 analyzer with cells sequentially treated with glucose (10 mM), oligomycin (1.0 µM) and 2-deoxyglucose (2-DG 50 mM).

The fatty acid palmitate-oxidation stress test was performed as per the manufacturer’s instructions (Agilent XF Palmitate Oxidation Stress Test—Advanced Assay), except cells were not serum-starved due to constraints of in vivo stimulated primary cells. For cell media, XF DMEM medium ph 7.4 (Agilent) supplemented with 0.5 mM L-carnitine and 2.0 mM glucose was used. Cells were incubated at 37 °C for 15 min, then just prior to beginning the assay either Palmitate:BSA substrate (Agilent XF Palmitate-BSA FAO Substrate, 30 µL per well) or BSA control substrate (Agilent XF BSA Control FAO Substrate, 30 µL per well) were added to cells. OCR was measured on the XFe96 analyzer with cells sequentially treated with etomoxir (30 µM) or media, oligomycin (1 µM), FCCP (5 µM) and a combination of antimycin A (1 µM) and rotenone (1 µM).

All assays were performed in 9 min cycles of mix (3 min), wait (3 min) and measure (3 min) as per manufacturer’s recommendations. Data were averaged from 2–4 technical replicate wells and cellular metabolism readouts were determined using Agilent Seahorse Wave software.

### 2.7. RNA-seq

For granulosa cell RNA-seq, female PRKO, PRAKO and PRBKO mice at 21–27 days old were hormonally stimulated with 5 IU eCG followed by 5 IU hCG and were humanely killed at 8 h post-hCG. Granulosa cells were collected from ovaries by follicle puncturing, then snap-frozen in liquid nitrogen. A total of 4 biological replicates per genotype were obtained, each consisting of granulosa cells pooled from 3 mice. RNA was extracted with the RNeasy Mini Kit (Qiagen), including DNase treatment, as per the manufacturer’s instructions. RNA quality was assessed using the RNA ScreenTape system (Agilent). A total RNA library was prepared using the Universal RNA-seq kit with NuQuant Mouse AnyDeplete (Nugen, Redwood City, CA, USA). Paired-end sequencing was performed on the NovaSeq 6000 S1 sequencing system (Illumina) at a depth of 81–99 M, 100 bps paired reads. Overrepresented adapters were checked using FastQC [45] and AdapterRemoval [46] was used to trim when required. Reads were aligned to the GENCODE mouse transcriptome (GRCm38.p6 M25 release) and transcripts quantified using Salmon [47]. DEGs were assessed using the limma-voom method [48]. Differential gene expression was defined as logFC ≥ 1 (fold-change ≥ ±2) and a Benjamini–Hochberg adjusted *p*–value ≤ 0.01.

Stroma RNA-seq was performed on granulosa cell-depleted ovarian tissue, which includes primarily stromal cells but also vascular endothelial cells, theca cells and immune cells (collectively referred to as stromal tissue) [23]. Female PRKO mice at 21–27 days old were hormonally stimulated as above and ovaries collected at 8 h post-hCG. Ovaries were repeatedly and thoroughly punctured to remove granulosa cells and the residual ovary was snap frozen in liquid nitrogen. A total of 3–4 biological replicates per genotype were obtained, each consisting of tissue pooled from 3 mice. RNA was extracted from the unprocessed residual ovarian tissue using the RNeasy Mini Kit (Qiagen), including DNase treatment, as per the manufacturer’s instructions. RNA quality assessment, library preparation and sequencing were conducted by the Australian Genome Research Facility (AGRF, Melbourne, Australia). mRNA libraries were prepared using the Illumina Truseq Stranded mRNA kit. Paired-end, 100 bp sequencing was performed on a NovaSeq 600 SP, 200-cycle (Illumina) sequencing system with a depth of 54–72 M reads per sample. Sequencing quality was checked using FastQC. Transcripts were aligned to the mouse transcriptome (GRCm39 M26 release) and quantified with Salmon [47]. DESeq2 [49] was used to assess differentially expressed genes between KO and WT with differential gene expression defined as logFC ≥ 0.5 and a Benjamini–Hochberg adjusted *p*-value ≤ 0.05. To account for granulosa cell contamination in starting material, stroma DEG whose expression level in granulosa cells (GC) was at least double the expression level in stroma (count_(GC)/_count_(stroma)_ ≥ 2 or logFC_(GC)/_logFC_(stroma)_ ≥ 2) were excluded. From this, 83.2% of identified stroma DEGs were retained.

In silico pathway analysis of DEGs was performed through Ingenuity Pathways Analysis (IPA) software (QIAGEN, Hilden, Germany). Functions and disease and upstream regulators analyses were used, and the top dysregulated categories identified based on the highest absolute activation z-scores. The visualization of the DEG analysis data was conducted primarily using ggplot2 [50] and read alignments visualized using first HISAT2 [51] then the UCSC Genome Browser (Santa Cruz, CA, USA). Raw data from RNA-seq are available on GEO (GC-GSE168213/Stroma-GSE197759)

### 2.8. Quantitative Real-Time PCR

Granulosa cells were collected from ovaries by repeated puncturing. RNA was isolated using the TRIzol extraction method, including DNase treatment. cDNA was then synthesized from 1100 ng of RNA using SuperScriptIII reverse transcriptase kit (Thermo Fisher Scientific, Waltham, MA, USA). cDNA was used for RT–qPCR using TaqMan assay probes and methodology (Thermo Fisher Scientific). The expression was normalized to the Rpl19 reference gene and expressed as fold change relative to the mean of the control samples using the ddCT method. Each group had 4 biological replicates (each pooled from 3 mice). Taqman probes: Zbtb16 Mm01176868_m1; Pparg Mm01184322_m1; Rpl19 Mm02601633_g1.

### 2.9. Western Blot

Granulosa cells were collected at 10 h post hCG by ovary puncture. Cells were lysed in RIPA buffer (Merck, Darmstadt, Germany) containing a 1% protease inhibitor cocktail (Sigma-Aldrich). Proteins were denatured in LDS buffer containing 1 µL β-Mercaptoethanol at 65 °C for 10 min, for ZBTB16 blot, or 50 °C for 10 min for OXPHOS blot. Equal quantities of protein were loaded into 4–12% Bis-Tris gel, and proteins separated by gel electrophoresis at 165 V for 45–60 min. Protein was transferred onto nitrocellulose membrane and then blocked in Odyssey blocking buffer (Li-Cor, Lincoln, NE, USA) for 1 h at RT. Blots were incubated overnight at 4 °C with primary antibodies (OXPHOS (1:1000, ab110413, Abcam, Cambridge, UK) anti-ZBTB16 (1:1000, AF2944, R&D systems, Minneapolis, MN, USA) or anti-beta actin (1:5000, A1978, Sigma-Aldrich, Burlington, MA, USA). After washing 3 × 5 min in PBS-0.1% Tween, membranes were incubated with corresponding secondary antibodies (Li-Cor) for 1 h at RT, then washed again. Membranes were imaged on the Odyssey imager infrared imaging system (Li-Cor). Western quantification is shown as fold change relative to reference protein, B-actin. We used 3 biological replicates (pooled from 4 mice) per genotype.

### 2.10. Statistical Analysis

The statistical analysis of the data was performed using Graphpad Prism 9, Sigmaplot or R software (https://www.r-project.org, accessed on 11 March 2022). All experiments involved multiple biological replicates and data points were from ovaries of individual mice or cells pooled from multiple mice as indicated. The statistical analysis of the RNA-seq DEGs was performed as stated above. All bar graphs are shown as mean ± SEM. Comparisons between control and KO were analyzed by unpaired *t*-test. Comparisons between WT, het and KO or over a time course were analyzed using a one-way ANOVA with Tukey’s multiple comparisons. * *p* < 0.05, ** *p* < 0.01, *** *p* < 0.001.

## 3. Results

### 3.1. PGR Specifically Regulates Rupture of the Follicle at Ovulation

The progesterone receptor (PGR) was highly abundant in granulosa cells of ovulating follicles, as shown by immunofluorescence staining of wildtype mouse ovaries at 6 h post hCG-stimulation. Within the follicle, the PGR was localized specifically to the granulosa cells, with no PGR observed in the cumulus cells or oocyte (Figure 1A). In addition, a population of cells within the ovarian stroma expressed the PGR at 6 h post-hCG (Figure 1B). This staining pattern was validated with a second anti-PGR antibody (data not shown).

To investigate ovulatory mechanisms controlled by the PGR, PRKO females and heterozygous littermate controls were hormonally stimulated with gonadotropins and ovaries collected along a time course. In mice, ovulation occurs at approximately 12–13 h post-hCG [5], therefore ovaries collected every 2 h from 8–16 h post-hCG were histologically examined for classical periovulatory features (Figure 2). At 8 h, the COCs were still relatively compact, but by 10 h post-hCG, a similar COC expansion level had occurred in the ovaries in both genotypes. The vascularization of the thecal layer around antral follicles and large blood vessels in the stroma was evident for both PRKO and control ovaries. By 12 h post-hCG, there was a significant COC expansion, and no apparent differences in COC expansion in the PRKO ovaries. In addition, an accumulation of the granulosa cells at the follicle base and a substantial thinning of the apical wall occurred in PRKO ovaries to a similar degree as the controls. Therefore, through histological examination, there were no morphological differences in the PRKO ovaries and classical ovulatory mechanisms of cumulus cell expansion and follicle remodeling appeared histologically normal.

At 14 h, a striking difference between PRKO and control ovaries became apparent, with a clear anovulation phenotype in the PRKOs. Control follicles had formed corpora lutea with confluent cell layers and no remaining antral cavity, indicating rupture and oocyte release had occurred. Conversely, the PRKO ovaries had persistent antral cavities with the oocytes and expanded cumulus masses remaining trapped within the follicles. At 16 h post-hCG, there were still multiple unovulated follicles within the PRKO ovaries. Some follicles had persistent antral cavities with trapped oocytes and other follicles had dark confluent cells surrounding a trapped oocyte indicating luteinized, unruptured follicles. From this histological time course, it appeared that the PGR very specifically regulated the rupturing of the follicle; however, its PGR-regulated downstream pathways were not clear. Therefore, the mechanisms behind PGR regulation of follicle rupture required a closer functional investigation.

### 3.2. Ovarian Progesterone Receptor Is Essential to Ovulation and Fertility

Ovarian transplantation experiments were conducted to confirm that ovarian expression of the PGR was solely responsible for the anovulation phenotype. PRKO or WT ovaries were transplanted into the contralateral sides of recipient ovariectomized WT mice (Figure 3A). After superovulation, the KO ovaries transplanted in WT recipients contained unovulated follicles (Figure 3B). There was a significantly higher number of anovulatory follicles in the transplanted KO ovaries, with an average of 23 (±2.33) compared to 4 (±1.86) in the transplanted WT control ovaries (Figure 3C). This abundance and the appearance of the unruptured follicles in the transplanted PRKO ovaries is analogous to what is seen in mice with a global PGR-deletion.

Additionally, the fertility of ovariectomized mice with either a single KO or WT ovary transplantation was examined (Figure 3D). Females were co-housed with WT males with evidence of copulation observed between 1 and 7 days after pairing in all females including those with KO ovaries, demonstrating that mating receptivity behavior did not require ovarian PGR signaling. Two of the three females with a WT ovary transplant produced pups totaling 14 offspring over five litters for WT ovary recipients. There were no pups or fetuses produced from the KO ovaries with no offspring produced during mating with a PR+/− genotype, indicating that fertility could not be restored to the PRKO ovaries even when transplanted to a PGR-replete recipient (Figure 3E). This shows that ovarian expression of the PGR is essential for follicle rupture and oocyte release. Therefore, it is essential to understand the mechanisms regulated by PGR within the ovary.

### 3.3. Ovulatory Cumulus–Oocyte Complexes Are Subtly Altered in PRKO Mice

The PGR was not expressed in the cumulus–oocyte-complex (Figure 1A); however, it was investigated if the PGR had any downstream effects on important cumulus cell function, including expansion and oocyte maturation. The COCs isolated at 10 h post-hCG showed no morphological differences between PRKO and heterozygous littermates in COC expansion or integrity (Figure 4A), consistent with tissue histology. An RNA microarray of the PRKO COCs was used to identify any transcriptional differences caused by PGR-deletion. At 8 h post-hCG, there were 73 differentially expressed genes (DEGs) in the PRKO COCs vs. the heterozygous control COCs, with 6 genes upregulated and 67 downregulated (logFC ≥ 1, padj ≤ 0.05) (Figure 4B, Appendix A). This suggests that the PGR may regulate transcriptional induction of some cumulus cell genes. However, when compared to an adjacent microarray dataset of the PRKO granulosa cells from the same mice, the majority of the COC DEGs were also dysregulated in the granulosa cell population [34,38]. Additionally, no cumulus matrix genes were dysregulated, which is consistent with observed normal COC expansion. Therefore, there was minimal PGR regulation of cumulus cell-specific gene expression.

Another important function of the cumulus cells is to support oocyte development. The effect of PRKO on oocyte development was assessed by analyzing mitochondrial bioenergetic activity in the oocyte, which has been linked to oocyte viability and developmental competence [52]. An image analysis of mitochondrial JC-1 staining showed a consistent pattern in control oocytes with high (red) mitochondrial potential concentrated to the pericortical region and low (green) mitochondrial potential predominant across the central cytoplasm (Figure 4C). Interestingly, while the PRKO green fluorescence intensity was similar to controls, there was a significant increase in red fluorescent staining across the center of the PRKO oocytes indicating more mitochondria with high membrane potential in the center of PKO oocytes (Figure 4D). Therefore, the distribution of mitochondria with high membrane potential was slightly skewed in the PRKO oocytes, potentially an indication of altered COC function or oocyte maturation. Importantly however, we have previously found that when oocytes of PRKO mice were matured in vitro and subjected to IVF, they gave rise to embryos that were indistinguishable from wildtypes, and following uterine transplantation produced healthy live offspring [53].

### 3.4. Mitochondrial Metabolism Is Not Altered in PRKO Granulosa Cells at Ovulation

Mitochondrial metabolism is a pathway regulated by PGR and other steroid receptors in certain biological contexts [54,55,56], and our analysis of LH-induced gene expression found mitochondrial transport pathways were key functions upregulated in granulosa cells at ovulation [39]. To determine if PGR also influences mitochondrial metabolic rate in granulosa cells during ovulation, the protein levels of electron transport chain components and the metabolic profile of ovulatory granulosa cells were assessed. No difference was seen in the abundance of mitochondrial oxidative phosphorylation subunit proteins in PRKOs compared to control littermates at 10 h post-hCG (Appendix A). All respiration function parameters including basal oxygen consumption rate (OCR), maximal OCR and ATP production were not significantly different between PRKO granulosa cells and controls (Figure 5A,B). Likewise, the glycolysis stress test found no difference in basal glycolysis or glycolytic capacity between cells from PRKO mice and controls (Figure 5C,D). Mitochondrial respiration and glycolysis were similarly not affected in granulosa cells from PRAKO or PRBKO mice (Appendix A).

Fatty acid oxidation stress test results found no difference in fatty-acid oxidation in the PRKO cells. The addition of etomoxir (acute response) reduced respiration capacity in all groups, demonstrating that fatty acids were used as an energy source in normal ovulatory granulosa cells; however, there were no differences detected between KO, WT or heterozygous cells. There was no change in endogenous maximal response, suggesting no effect of the PGR on the utilization of intracellular fatty acids nor was there a difference in the utilization of palmitate at basal or maximal response conditions (Figure 5E,F). Thus, although mitochondrial respiration, glycolysis and fatty acid oxidation are evident metabolic processes occurring in granulosa cells immediately prior to ovulation, no changes were seen in these metabolic capacities due to the loss of the PGR. Therefore, the PGR does not appear to regulate ovulation through overarching changes in granulosa cell energy production.

### 3.5. Key PGR Downstream Pathways Include a Transcription Factor Network and Actin Fiber Gene Regulation

To determine the PGR isoform-specific preovulatory transcriptome in granulosa cells, RNA-seq was performed on granulosa cells from PRKO, PRAKO and PRBKO mice [39]. Granulosa cells were isolated at 8 h post-hCG and differentially expressed genes (DEGs) determined between KO and WT controls. The principal component analysis is shown in Appendix A. The loss of both PGR isoforms (PRKO) resulted in 236 DEGs (logFC ≥ 1, padj ≤ 0.01). The loss of only the A isoform (PRAKO) resulted in 310 DEGs and of these, 154 were common with the PRKO DEGs (Figure 6A). Conversely, the loss of PGR-B (PRBKO) showed no significant DEGs (logFC ≥ 1, padj ≤ 0.01). This is demonstrated on the gene level across *Cxcr4* with PRKO and PRAKO showing a decreased transcript level while PRBKO was equivalent to that of WT (Figure 6B). This indicates that PGR-A is the predominant isoform regulating transcription in preovulatory granulosa cells.

As PRKO and PRAKO mice both have deficient ovulation, the genes most likely to be essential for follicle rupture are those that are dysregulated in both of these models. Thus, the list of 154 common DEGs was interrogated further in order to specifically investigate ovulatory function (Appendix A). A pathways analysis was applied to classify PGR-regulated gene networks and identify potential downstream functions. An IPA function and disease analysis (Figure 6C) predicted that the loss of PGR led to a decrease (negative activation z-score) in three main function groups: actin cytoskeleton dynamics, invasion of tumor cells and quantity of cells (groups determined manually). The most highly deactivated function was “formation of actin filaments” (z = −2.6, *p* = 0.0007) followed closely by “formation of actin stress fibers” (z = −2.4, *p* = 0.0017). Invasion of tumor cell categories include “growth of tumor” (z = −2.0, *p* = 0.00003) and “invasion of cells” (z = −1.711, *p* = 0.00005). While the third collection of dysregulated functions, “quantity of cells”, includes “proliferation of epithelial cells” (z = −2.1, *p* = 0.00001) and “quantity of leukocytes” (z = −2.0, 0.00003). The DEGs associated with the most highly dysregulated function, “formation of actin filaments” included *Cdk6* (PRKO logFC = −2.34, padj = 0.00008), *Pdlim4* (PRKO logFC = −1.77, padj = 0.00067) and *Sorbs3* (PRKO logFC = −1.74, padj = 0.0041) (Figure 6D).

Interestingly among the 154 common DEGs, 11 genes (9 downregulated, 2 upregulated in KO) were transcription factors (TFs) (Figure 7A). This indicated that during the periovulatory window, the PGR was orchestrating a secondary level of transcriptional regulation by activating DNA-binding TFs including *Zbtb16* (PRKO logFC = −4.05, padj = 0.00016) and *Epas1* (PRKO logFC = −2.6, padj = 0.00011) and transcription cofactors such as *Cited1* (PRKO logFC = −2.77, padj = 0.0095). The expression of transcription factors *Pparg* and *Zbtb16* was LH-induced by 14- and 30-fold, respectively (Figure 7B) and PGR regulation of *Pparg* and *Zbtb16* was confirmed with qPCR (Figure 7C). *Pparg* was downregulated in PRKO and PRAKO but not PRBKO granulosa cells. The transcriptional regulation of *Zbtb16* was also established with a striking block in expression in PRKO granulosa cells (33-fold decrease to WT) and partial inactivation seen in the heterozygous littermates. The PRAKO cells showed a very similar transcriptional dysregulation of *Zbtb16* while the PRBKO cells also exhibited a decrease in *Zbtb16* mRNA (threefold decrease to WT). The reduced abundance of ZBTB16 protein in granulosa cells was also confirmed at 10 h post-hCG. A Western blot demonstrated a decrease in protein in the PRKO and PRAKO granulosa cells while there was no significant decrease in ZBTB16 protein in PRBKOs (Figure 7D,E).

### 3.6. PGR Regulation of the Ovarian Stromal Transcriptome

RNA-seq analysis was performed for PRKO ovarian stromal tissues collected under the same conditions as granulosa cells. A principal component and differential gene expression analysis show that the loss of the PGR also had some effect on the stroma, though there was a lower extent of impact on the magnitude of the differential expression in the stroma compared to the granulosa cells (Appendix A). Nevertheless, 104 stromal DEGs were identified, with 86 upregulated and 18 downregulated, using parameters logFC ≥ 0.5, padj ≤ 0.05 (Figure 8A, Appendix A). The DEGs identified included the lipid coating gene *Plin4* (logFC = −1.68 padj = 0.0002), phospholipase inflammatory enzyme *Pla2g5* (logFC = −1.51, padj = 0.000002) and vasoconstriction factor *Edn2* (logFC = −1.29 padj = 0.0000001) with a difference in mRNA expression demonstrated on read count histograms showing clear exon peaks for the WT replicate compared to the KO (Figure 8B). As determined by the IPA functions and disease analysis (Figure 8C), the most highly downregulated function caused by the loss of the PGR was “synthesis of lipid” (z = − 3.1, *p* = 0.0009), with over half of the dysregulated functions falling into this lipid metabolism broad category including deactivation of “synthesis of steroid” (z = −2.2, *p* = 0.0008). Other highly affected functions include “phagocytosis of cells” (z = −2.2, *p* = 0.0007) and “engulfment of cells” (z = −2.4, *p* = 0.0027).

To further examine the PGR-regulated networks within the stroma, the upstream regulators of the stromal DEG genes were identified. The top 20 upstream regulators in the stroma were then compared to the upstream regulators identified in the granulosa cell RNA-seq to determine if PGR was regulating unique networks in each cell type (Figure 9A). Interestingly, the PGR-regulated networks in the stromal cells were rather distinct from those in the granulosa cells. Of the 20 predicted deactivated upstream regulators identified in the stromal cells, only 4 of these were also predicted to be deactivated upstream regulators in the granulosa cells. The common upstream regulators included IL2, Ca2+, dexamethasone and beta-estradiol. In contrast, the unique stromal upstream regulators included cholesterol synthesis transcription factors SREBF2 and SREBF1. Surprisingly, the INHA, AGT, PPARG and TNF networks were predicted to be deactivated due to the PGR deletion in stromal tissue, but the same upstream regulators were oppositely active in the granulosa cells. This suggests that PGR regulates distinct stromal-specific gene networks.

## 4. Discussion

This study demonstrated that the PGR in the ovary alone was responsible for initiating the necessary changes that lead to follicular rupture. Further, we showed that the PGR, and in particular PGR-A, regulated a broad spectrum of genes with diverse functions in the granulosa cells at ovulation. Specifically, this study identified a secondary network of transcription factors that are PGR-regulated and may be orchestrating downstream ovulatory pathways. Despite no histological changes in the PRKO ovaries, we employed pathways analysis to identify novel function mechanisms that may be required for ovulation. In doing so, a link between PGR and actin fiber dynamics and cell invasion was identified. In addition, the function of the stroma at ovulation was considered, finding that lipid processing genes were the key stromal networks affected by the loss of the PGR. The PGR-dependent pathways identified in this study were summarized in Figure 9B.

Fundamentally, the ovarian transplant of PRKO ovaries into WT recipients demonstrated that the PGR expression within the ovary was critical to follicular rupture and that a normal PGR expression in the brain, pituitary and other reproductive tissues was not able to rescue the infertility caused as a result of the PGR deletion in the ovary. This finding is supported by another recent study using gonadotrope-specific PGR conditional KO mice to show that despite a blunted LH-surge, gonadotrope-KO females otherwise exhibited a normal estrus cycle and normal fertility [32]. Similarly, a global PGR deletion leads to altered sexual receptivity behavior [24]. The transplant experiments demonstrated that this lack of hypothalamic PGR expression did not influence ovulation or luteal maintenance since WT ovaries transferred to a KO recipient exhibited ovulated follicles and supported pregnancies. Together these studies demonstrate that PGR expression in the ovary specifically is a requisite for ovulatory function.

Within the ovary, our immunostaining confirmed an LH-induced PGR expression in the granulosa cells but not cumulus cells and also clearly showed the PGR expression in a population of stromal cells. We further demonstrated the PGR regulation of both the stromal and granulosa cell transcriptome. Additionally, potentially through the regulation of downstream intrafollicular signaling networks, the PGR mediated some changes in the cumulus–oocyte complex including a disruption of the oocyte mitochondrial distribution. However, the PGR expression was not detected in oocytes (Figure 1) and it does not appear to be essential for oocyte developmental competence [53]. The difference in mitochondrial activity is therefore most likely due to alterations in PR-mediated granulosa cell responses subtly influencing the oocyte microenvironment. Mitochondrial metabolism was more extensively examined in granulosa cells, but a lack of PGR did not significantly change oxygen consumption, glycolysis or fatty acid oxidation. Thus, using this model, we cannot infer any conclusions about whether a disruption of these metabolic processes would influence ovulatory capacity. Overall, an extensive histological analysis of the ovary showed no obvious disruption of the known physiological changes necessary for ovulation, apart from follicle rupture indicating that the PGR may be regulating ovulation through unknown mechanisms.

To assess novel pathways by which the PGR may regulate ovulation, we employed an RNA-seq analysis and discovered that in the granulosa cells, the PGR regulated genes associated with actin cytoskeleton dynamics, such as both *Pdlim1* and *Pdlim4* scaffolding proteins, which are required for actin stress fiber formation [57,58]. The indication of granulosa cell actin dynamics having a conserved role in ovulation is supported by a recent microarray analysis of human mural granulosa cells, in which “actin cytoskeleton signaling” was one of the top enriched canonical pathways with predicted activation at ovulation (32–36 h poststimulation) [59]. Other key PGR-regulated genes were associated with the invasion of cells such as *Cxcr4* and *Cldn1*, two membrane-localized proteins involved in tumor cell invasion and migration [60,61]. The invasion of cells is a known function for cumulus cells at ovulation, as cumulus cells gain a significant invasive, adhesive and migratory capacity in response to the LH surge, peaking at the point of ovulation [5]. Future analysis to identify the individual or combinatorial genes critical for ovulation, and also a further investigation of additional transcriptional mechanisms such as PGR-regulated noncoding-RNAs, could provide new, ovulation-specific targets for fertility therapeutics.

The PGR was also found to regulate a secondary network of transcription factors during ovulation. One of the most dysregulated genes in the PRKO granulosa cells was in the LH-induced transcription factor *Zbtb16*. While both PGR isoforms have a degree of influence on *Zbtb16* transcription, its expression was more dependent on the PGR-A isoform, consistent with our current and previous finding, that PGR-A is the more transcriptionally active isoform at ovulation [39]. While ZBTB16 action in the ovary remains unknown, it is known to mediate transcriptional changes downstream of PGR such as in the uterus, where PGR-induced ZBTB16 is important for mediating genes involved in progesterone-dependent decidualization [62]. PPARγ is another transcription factor dependent on the PGR, considered critical in ovulation and known to regulate ovulatory genes *Edn2* and *Prkg2* in granulosa cells [37]. We identified that the PGR-A isoform alone was responsible for the induction of *Pparg* as it was similarly dysregulated in the PRAKO and the total PRKO granulosa cells. Interestingly, *Pparg* was also PGR-regulated in the stroma and identified as an upstream regulator of stromal DEGs, suggesting an additional role for PPARγ in the stroma, potentially in the regulation of lipid metabolic pathways [63]. It is evident that the PGR is orchestrating a multileveled network of transcriptional changes in granulosa cells and that these may be inducing changes within the ovarian stromal tissue.

We found that the PGR was expressed within the ovarian stromal compartment. There were many differentially regulated genes identified in the stroma, however, these had a lower fold-change compared to the granulosa cell DEGs confirming that the PGR exerts more dramatic changes within the granulosa cell population. This was expected as the PGR is highly induced in the granulosa cells following the LH-surge. Moreover, the stroma contains endothelial, immune and theca cells among others, potentially diluting the magnitude of changes that can be detected in underrepresented cell populations. There were no protease or remodeling DEGs detected in the stromal tissue, suggesting that PGR does not affect stroma-controlled follicle remodeling for ovulation, at least at the transcriptional level.

A pathways analysis suggested that the PGR affects lipid synthesis pathways in the broad stromal tissue. This may be due to the PGR regulation of theca cell steroid synthesis or the regulation of theca to luteal differentiation. Additionally, “phagocytosis of cells” was a top regulated pathway associated with inflammatory genes such as *Pla2g5,* a phospholipase involved in inducing inflammatory response and promoting phagocytosis in leukocytes. Multiple cytokines and immune regulators such as IL4, TNFSF11, IL6, IL2 and TNF were also predicted as upstream regulators of stromal DEGs suggesting that the PGR may be stimulating an inflammatory response in the stroma, a potential mechanism important for ovulation. This finding is supported by our previous low-density array analysis of whole ovary samples collected at 10 h post-hCG, which identified an altered expression of inflammatory mediators in PRKO mice [38]. However, another study using scRNA-seq on granulosa conditional PGR-KO mice had conflicting conclusions, proposing that the PGR acts as an inflammatory suppressor during ovulation [64]. It is important to note, however, that the granulosa cells were assessed at an earlier ovulatory timepoint (6 h post-hCG), compared to the current study, which analyzed tissue collected at 8 h post-hCG. As the periovulatory window is considerably shorter (12 h) and involves the activation of extensive cellular and morphological changes within the follicle, the genes that are vital to ovulation are likely to have highly precise expression patterns. For example, while at 6 h post-hCG *Ptgs2* was found to be attenuated by PGR in granulosa cells, at 8 h post-hCG such effect was no longer observed [38]. Therefore, our conclusions support that closer to the time of ovulation, the PGR may have a role in stimulating the inflammatory response.

Overall, we conclude that the PGR expression in the ovary alone is necessary for follicular rupture. We showed that PGR-A was the main regulator of the ovulatory granulosa cell transcriptome and was especially important as an upstream factor of a transcription regulatory network through the activation of transcription factors such as *Zbtb16* and *Pparg*. One or multiple of the pathways uncovered may be the PGR-directed downstream mechanism essential for follicle rupture and identifying these critical mechanisms will provide new targets and insights into the development of fertility therapies and non-hormonal contraceptives.

## Figures and Tables

**Figure 1 cells-11-01563-f001:**
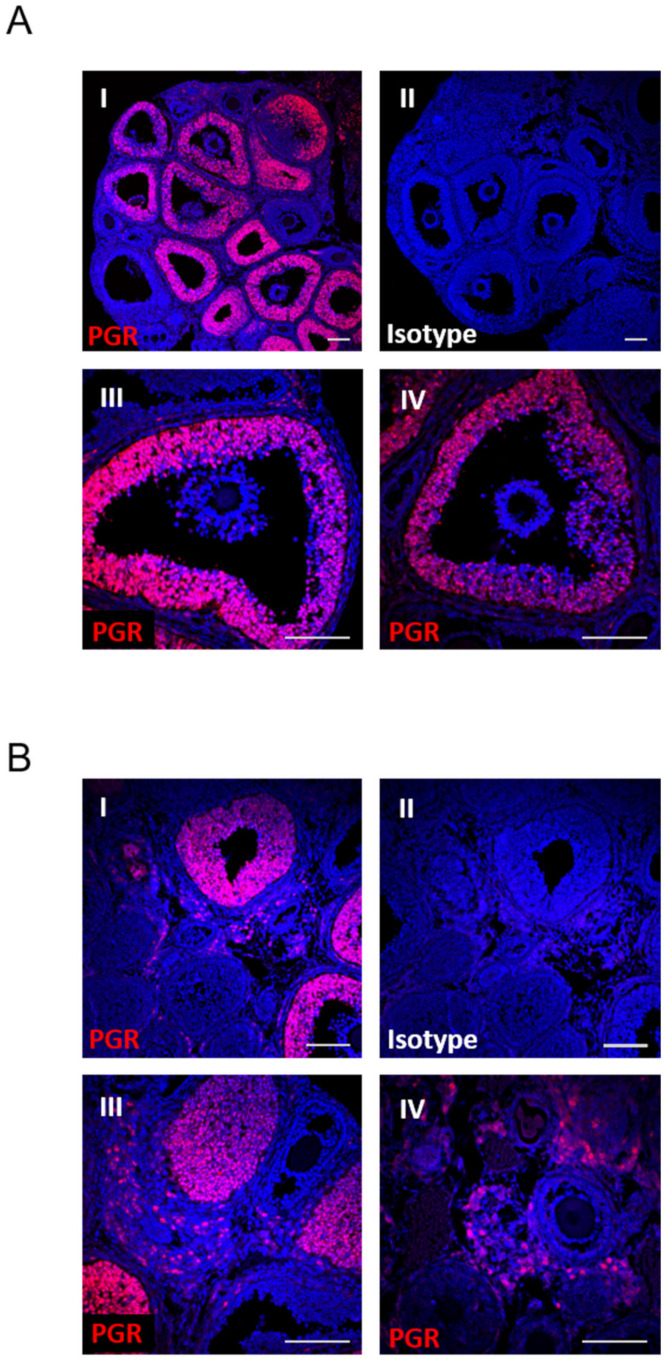
PGR is expressed in preovulatory granulosa cells and within the stromal compartment but not in cumulus cells. PGR localization by immunohistochemistry in wildtype ovaries at 6 h post-hCG using anti-PGR antibody (red) and Hoechst33342 nuclear stain (blue). (**A**) Ovary sections stained with anti-PGR (I) or isotype control (II)), with individual follicles shown at higher magnification (III, IV). (**B**) Ovary sections stained with anti-PGR (I) (or isotype control (II)), with the stromal compartment shown at higher magnification (III, IV). All scale bars are 100 µm.

**Figure 2 cells-11-01563-f002:**
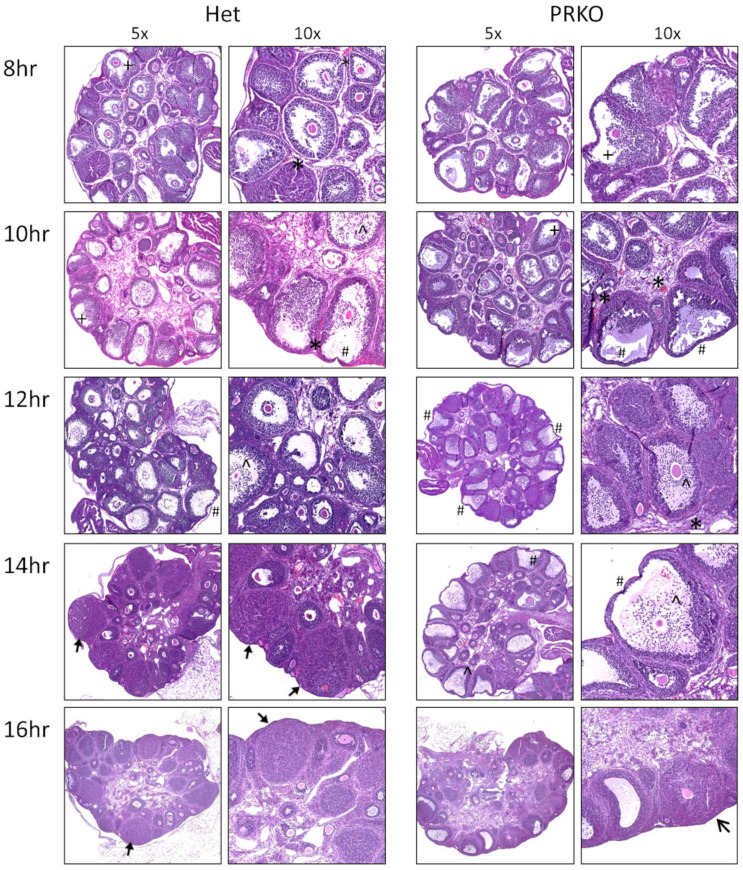
Histological analysis of PRKO ovaries demonstrates specific defect in follicular rupture. Ovaries were collected from PRKO and heterozygous (Het) littermates at 8, 10, 12 h timepoints post-hCG and 14 h, and 16 h timepoints, after ovulation has normally occurred. Ovary sections were stained with H&E to analyze histological phenotypes including ^ COC expansion, + granulosa cell accumulation at the follicle base, # thinning of the apical wall, * vascularization, luteinization, unruptured luteinized follicles. Representative examples from histology performed on ovaries from 3 mice per timepoint.

**Figure 3 cells-11-01563-f003:**
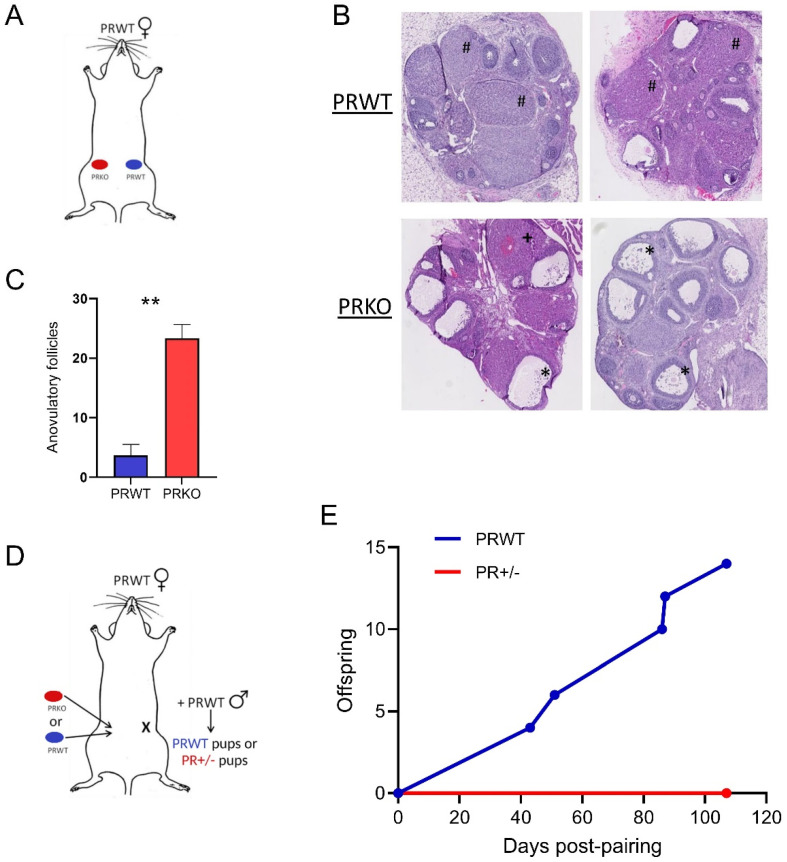
Ovarian progesterone receptor is essential for fertility. (**A**) Schematic of ovarian transplant design for histological assessment. Donor WT or PRKO ovaries were transplanted into the contralateral sides of recipient ovariectomized WT females. (**B**) Representative H&E images of transplanted WT and KO ovaries at 23 h post-hCG. Symbols represent follicle types: # normal luteinized follicles; * anovulatory unruptured follicles with expanded COCs; + anovulatory luteinized, unruptured follicles with entrapped oocytes. (**C**) Quantification of anovulatory follicles in KO vs. WT transplanted ovaries. Mean ± SEM; *n* = 3 donor ovaries per genotype. ** *p* < 0.01 by unpaired *t*-test. (**D**) Schematic of ovarian transplant design for fertility assessment. Ovariectomized recipient WT females were transplanted with either one WT ovary or one KO ovary then paired with WT stud males. (**E**) Offspring originating from a WT ovary (offspring PRWT) or KO ovary (offspring PR+/−) were tallied with numbers indicating total from litters of *n* = 3 transplanted recipients per genotype.

**Figure 4 cells-11-01563-f004:**
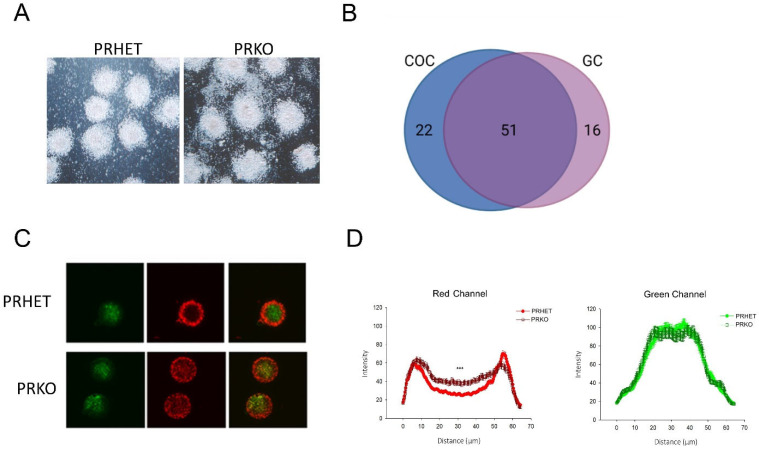
PRKO mice have altered mitochondrial distribution and changes in cumulus cell gene expression. (**A**) COCs collected from antral follicles of PRKO and heterozygous controls at 10 h post-hCG and photomicrograph images captured. (**B**) Microarray analysis was performed on COCs at 8 h post-hCG and PRKO differentially expressed genes (DEGs) identified (logFC ≥ 1, padj ≤ 0.01). COC DEGs were compared to previous microarray data of DEGs in the granulosa cells (GCs). (**C**) Denuded oocytes from PRKO and heterozygous controls were stained with JC-1 potentiometric dye to determine high mitochondrial membrane potential (red) and low mitochondrial potential (green). (**D**) JC-1 red channel and green channel fluorescence intensity and distribution were analyzed across the oocyte midsection. *n* = 4 mice per genotype, with an analysis of 58–61 oocytes. Statistical significance between genotypes was determined by mixed-model two-way ANOVA, *** *p* < 0.001.

**Figure 5 cells-11-01563-f005:**
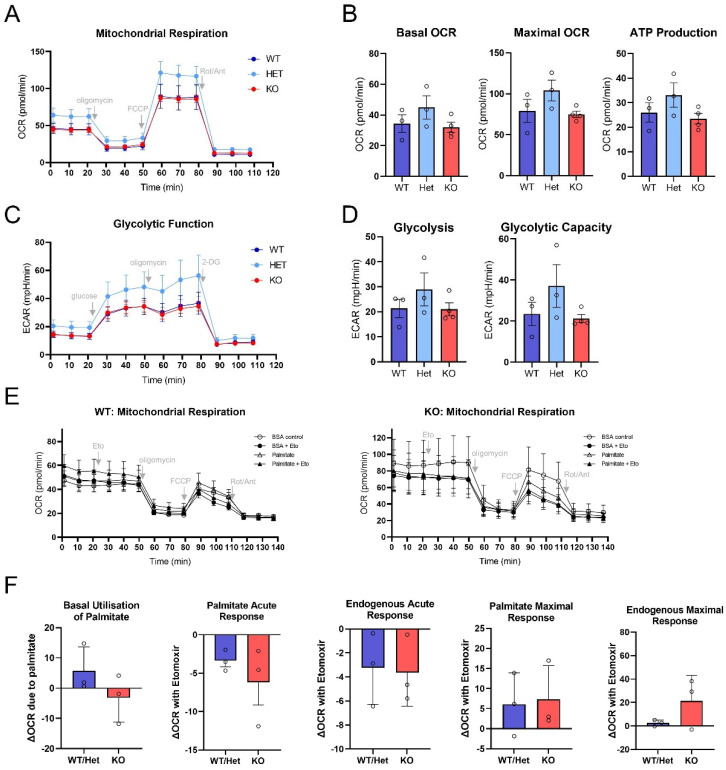
Granulosa cell mitochondrial metabolism, glycolysis and fatty acid oxidation are not altered in PGR null mice at ovulation. Granulosa cells were collected from PRKO mice and WT or HET littermates at 10 h post-hCG and assayed using Seahorse extracellular flux stress tests. (**A**,**B**) Mitochondrial stress test analysis of mitochondrial respiration capacity and ATP production. (**C**,**D**) Glycolysis stress test assessment of glycolytic capacity of granulosa cells. (**E**,**F**) Palmitate-oxidation stress test determined granulosa cell utilization of endogenous fatty acids or exogenous palmitate substrate for respiration. Values represent mean ± SEM of *n* = 3–4 biological replicates pooled from 1–3 mice each. No statistical differences were detected by one-way ANOVA (mitochondrial and glycolysis stress tests) or unpaired *t*-test (palmitate-oxidation stress test).

**Figure 6 cells-11-01563-f006:**
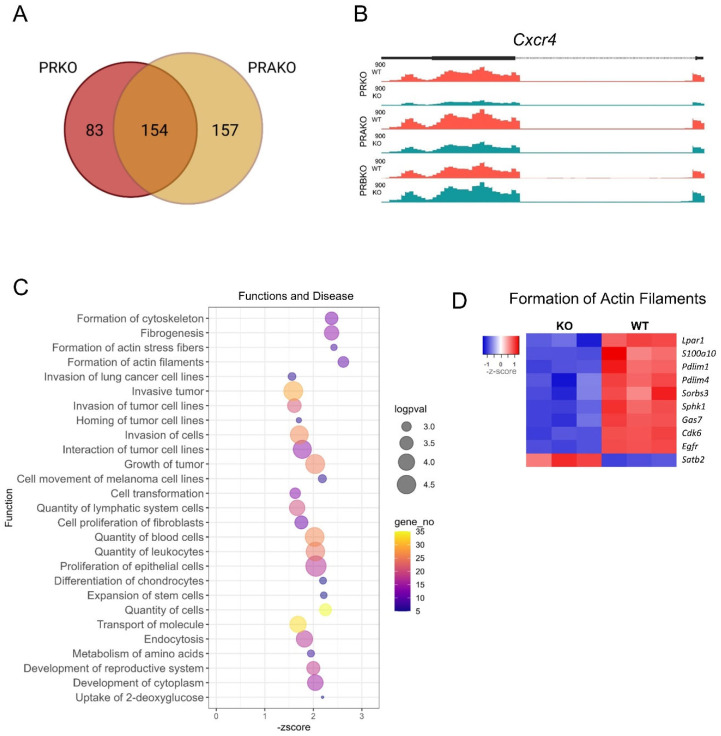
RNA-seq of PRKO, PRAKO and PRBKO granulosa cells identifies PGR-regulated genes and pathways. RNA-seq was performed on WT and KO granulosa cells of the PRKO, PRAKO and PRBKO strains. (**A**) RNA-seq analysis identified 236 DEGs in PRKO GCs and 310 for PRAKO GCs with 154 common DEGs (logFC ≥ 1, padj ≤ 0.01). (**B**) Example of mapped RNA-seq reads for PRKO, PRAKO and PRBKO depicting genome site for *Cxcr4*. (**C**) The 154 DEGs common to PRKO and PRAKO were analyzed in IPA and most highly downregulated function and diseases identified, depicted with log *p*-value (size), the number of DEGs associated with the function (color) and z-score indicating degree of downregulation (*X*-axis). Functions are grouped based on similarities. (**D**) Heatmap depicting the DEGs associated with “formation of actin filaments” function. RNA-seq reads per samples (log-counts per million, LCPM) are shown for PRKO (left) and PRWT (right) where blue is low and red is high expression.

**Figure 7 cells-11-01563-f007:**
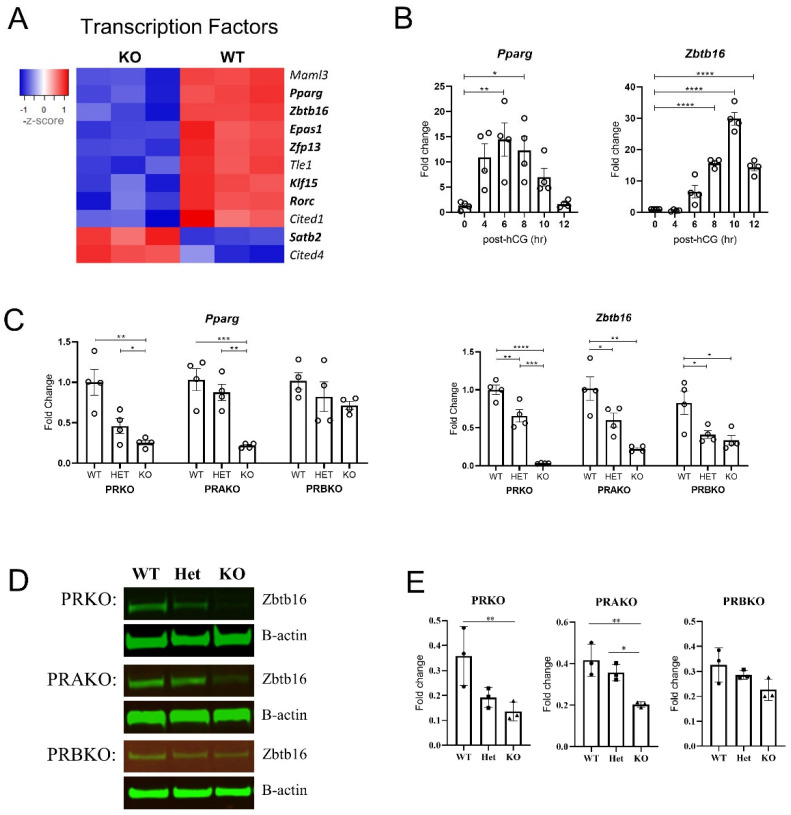
PGR regulates a network of transcription factors in granulosa cells including *Pparg* and *Zbtb16*. (**A**) Heatmap showing expression level of PGR-regulated transcription factors, including DNA-binding transcription factors (bold) and transcription cofactors (not bold). Shown are PRKO (left) and PRWT (right) RNA-seq reads (LCPM) with blue representing low expression levels and red increased expression levels. (**B**) Expression of *Pparg* and *Zbtb16* in wildtype granulosa cells over a preovulatory time course from 0–12 h post-hCG using CBA-F1 mice. (**C**) Expression of *Pparg* and *Zbtb16* in granulosa cells of PRKO (left) PRAKO (center) and PRBKO (right) mice at 8 h post-hCG qPCR data normalized to reference gene *Rpl19* and expressed as fold change relative to 0 h or WT control. N = 4 biological replicates of cells pooled from ovaries of 3 females. (**D**) Western blot for ZBTB16 protein in granulosa cells of PRKO, PRAKO and PRBKO mice at 10 h post-hCG. (**E**) Quantification of Western blot ZBTB16 expression relative to B-actin loading control. N = 3 pools of cells from 3 females. Mean ± SEM, statistical significance determined by one-way ANOVA. * *p* < 0.05, ** *p* < 0.01, *** *p* < 0.001, **** *p* < 0.0001.

**Figure 8 cells-11-01563-f008:**
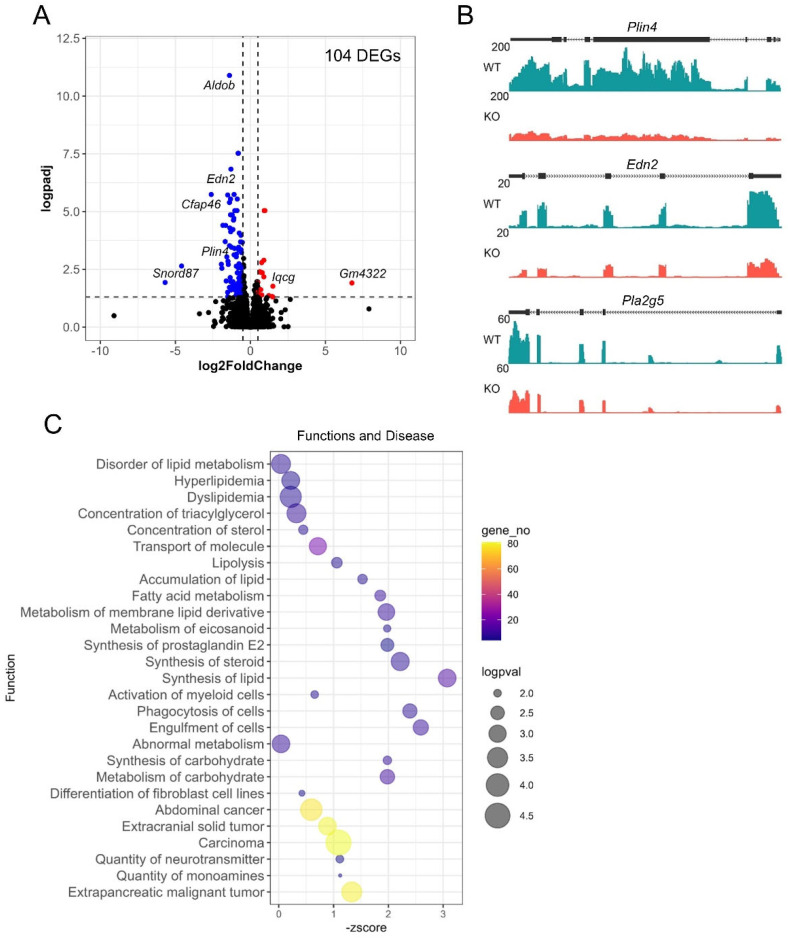
RNA-seq identifies PGR-regulation of stromal tissue. RNA-seq was performed on ovarian stromal tissue at 8 h post-hCG. (**A**) Volcano plot demonstrating the 104 differentially expressed genes (logFC ≥ 0.5 and padj ≤ 0.05) between PRKO and WT ovarian stromal tissue. Blue depicts downregulated and red depicts upregulated genes in PRKO stroma. (**B**) Example of *Plin4*, *Pla2g5* and *Edn2* differential gene expression as visualized by number of reads mapped on the gene. (**C**) “Function and diseases” dysregulated in PRKO ovarian stroma based on IPA analysis of stromal DEGs. Similar functions are grouped and shown as log *p*-value (size), the number of DEGs associated with the function (color) and z-score indicating degree of downregulation (*X*-axis).

**Figure 9 cells-11-01563-f009:**
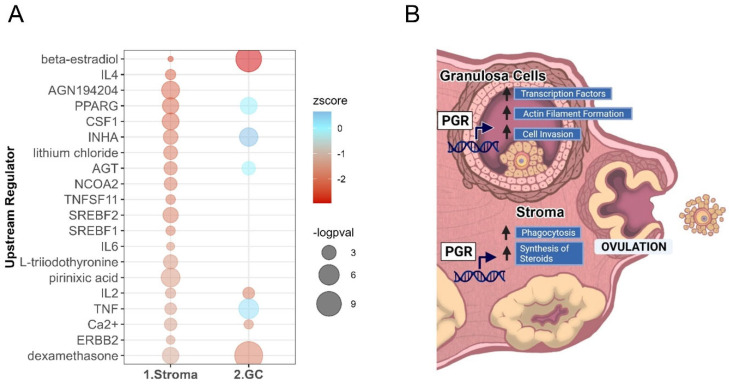
Pathways analysis comparing cell-specific DEGs shows unique regulation by PGR in stroma. (**A**) Top 20 deactivated upstream regulators predicted for stromal DEGs, ordered and colored by z-score indicating inactivation, sized by log *p*-val. The upstream regulators are also identified in the granulosa cell RNA-seq analysis shown on right. (**B**) Summary of the effect of PGR on intraovarian functional pathways at ovulation.

## Data Availability

The datasets generated and/or analyzed during the current study are available in GEO. Microarray analysis: COC—GSE92436, GC—GSE92437 and RNA-seq analysis: GC—GSE168213, Stroma—GSE197759.

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
