# Peer review of "Intraovarian, Isoform-Specific Transcriptional Roles of Progesterone Receptor in Ovulation"

_cells, 2022, doi:10.3390/cells11091563_

Round 1

Reviewer 1 Report

The manuscript presented a wide range of experiments provided to explain the PGR type-dependent coordination of gene expression involved in the ovulation process. Congratulations to the authors for conducting the complex research. The paper is well written and the results are presented accurately. There is only a lack of PCA analysis. The M&M sections are described in detail. 

Taking into consideration the results of the study, in the Discussion section Authors in a few places should clearly indicate the specific genes involved in actin filament formation and cells invasion in GCs due to this part of the Discussion appears flat.

Moreover, the Authors put special attention on Zbtb16 and Pparg in GCs cells and Pla2g5 in stromal tissue, therefore I wonder why, having the RNAseq results, they did not perform the single nucleotide variants (SNVs) analyses. Moreover, it should be interesting to perform the DE-ncRNAs analysis. It may be a suggestion to refill a conclusion section or directions of the further study.

Author Response

Reviewer 1
The manuscript presented a wide range of experiments provided to explain the PGR
type-dependent coordination of gene expression involved in the ovulation process.
Congratulations to the authors for conducting the complex research. The paper is well
written and the results are presented accurately. There is only a lack of PCA analysis.
The M&M sections are described in detail.

The PCA analysis of the RNA-seq data for each of the three granulosa cell
experiments and the stromal cell experiment is now shown in full as a new
Supplemental Figure 3.
Taking into consideration the results of the study, in the Discussion section Authors in
a few places should clearly indicate the specific genes involved in actin filament
formation and cells invasion in GCs due to this part of the Discussion appears flat.

Paragraph 4 of the Discussion is now updated to highlight some of the specific
genes involved in actin filament formation (Pdlim1, Pdlim4 ) and cell invasion
(Cxcr4, Cldn1) in granulosa cells.

Moreover, the Authors put special attention on Zbtb16 and Pparg in GCs cells and
Pla2g5 in stromal tissue, therefore I wonder why, having the RNAseq results, they did
not perform the single nucleotide variants (SNVs) analyses. Moreover, it should be
interesting to perform the DE-ncRNAs analysis. It may be a suggestion to refill a
conclusion section or directions of the further study.

We have now included this interesting suggestion to examine ncRNA in the
Discussion; however we have not mentioned SNV analysis as the RNA-seq
investigated PGR-regulated transcription of the normal genome and we were not
entirely sure of the link being proposed by the Reviewer.
New text, paragraph 4: Future analysis to identify the individual or combinatorial
genes critical for ovulation, and also further investigation of additional
transcriptional mechanisms such as PGR-regulated noncoding-RNAs, could
provide new, ovulation-specific targets for fertility therapeutics.

Reviewer 2 Report

1. In discussion section authors explain the role of PGR in the regulation of genes with important functions, such as transcription factors or actin cytoskeleton dynamics in granulosa cells or lipid synthesis pathways in stromal tissue. However, there are other results obtained that are not discussed, like the ovarian transplantation experiments, the higher mitochondria membrane potential in PKO oocytes or the metabolic capacities of granulosa cells. Maybe authors could include the discussion of all the results obtained.

2. In figure 2, what are arrows indicating? Please, complete Figure legend.

3. There are some not defined abbreviations such us PRHet or FAs.  

Author Response

Reviewer 2
In discussion section authors explain the role of PGR in the regulation of genes with
important functions, such as transcription factors or actin cytoskeleton dynamics in
granulosa cells or lipid synthesis pathways in stromal tissue. However, there are other
results obtained that are not discussed, like the ovarian transplantation experiments,
the higher mitochondria membrane potential in PKO oocytes or the metabolic
capacities of granulosa cells. Maybe authors could include the discussion of all the
results obtained.
We have added additional information to the Discussion as requested.
Specifically, each of the findings is summarized in full, including the ovarian
transplant experiments, the mitochondrial alterations in the PRKO oocytes and
granulosa cell metabolic assays. This information comprises two new
paragraphs (paragraphs 2 and 3) of the Discussion. In addition, further details
are provided about key DEGs of interest in paragraph 4 of the Discussion.
In figure 2, what are arrows indicating? Please, complete Figure legend.
The Figure 2 legend has been corrected to explain the arrows.
There are some not defined abbreviations such us PRHet or FAs.
Abbreviations are now defined.